# Assessing International Development Cooperation: Becoming Intentional about Unintended Effects

Dirk-Jan Koch [1,*] , Jolynde Vis [2], Maria van der Harst [2], Elric Tendron [2] and Joost de Laat [2,*]

[1] Department of Cultural Anthropology and Development Studies, Radboud Social Cultural Research and Radboud University, 6525 XZ Nijmegen, The Netherlands
[2] Utrecht University School of Economics, Centre for Global Challenges, Utrecht University, 3584 CS Utrecht, The Netherlands; j.t.vis@uu.nl (J.V.); a.m.r.vanderharst1@uu.nl (M.v.d.H.); elrictendron@gmail.com (E.T.)
[*] Correspondence: dirkjan.koch@ru.nl (D.-J.K.); j.j.delaat@uu.nl (J.d.L.)

**Abstract:** International headlines often make mention of side effects of international cooperation, ranging from aid-fuelled corruption to the negative side effects of volunteer tourism. The OECD Development Assistance Committee, an international forum of many of the largest providers of aid, prescribes that evaluators should consider if an intervention has unintended effects. Yet the little that is known suggests that few evaluations of international cooperation projects systematically assess their unintended effects. To address this gap in assessing unintended effects, this study develops an operational typology of 10 types of unintended effects of international cooperation that have emerged in the literature and applies this to all 644 evaluations of the Netherlands' development cooperation between 2000 and 2020 using structured text mining with manual verification. The results show that approximately 1 in 6 evaluations carefully considered unintended effects and identified 177 different ones. With the exception of 5, these could be classified in 9 of the 10 typologies, indicating that this typology can guide international development cooperation to systematically consider and assess its unintended effects. International development planners, researchers and evaluators are recommended to henceforth make use of and improve this operational typology.

**Keywords:** unintended effects; international cooperation; policy evaluation; typology of unintended effects





## 1. Introduction

International headlines occasionally make mention of the negative side effects of international cooperation, ranging from abusive behaviour by aid workers [1] to aid-fuelled corruption [2]. While there are also positive side effects of aid, such as deworming programmes leading to increased educational participation [3], these make it less often to the front pages. Because unintended effects can be sizeable, the OECD Development Assistance Committee, an international forum of many of the largest providers of aid, prescribes that evaluators should consider if an intervention has unintended effects. Yet the little that is known suggests that few evaluations of international cooperation projects systematically assess their unintended effects. Against this background, both negative and positive side effects of international cooperation have lately been attracting increased academic interest (for instance, Special Issues devoted to this topic in two journals, *Evaluation and Program Planning* and *The International Spectator*).

The existing literature provides important case studies focusing often on single policy areas (e.g., [4,5]). Abstract typologies of unintended effects have also been proposed (e.g., [6]), which enable reflection on the unintended effects of international cooperation in general terms. However, these heuristic devices have been less useful in guiding evaluators to identify actual unintended effects. Between the very specific case studies and the abstract typologies, a systematic theory-driven taxonomy of actual unintended effects is

lacking, and when researchers are asked to focus on unintended effects, the effort may be unsatisfactory, or researchers may even be at a loss as to what to look for (e.g., [7,8]). To be operationally useful, the taxonomy needs to focus on the actual mechanisms and processes that are set in motion by international cooperation and that contribute to unintended effects. A taxonomy should thus be centred around substantive and observed unintended changes in the field of international cooperation, instead of a more generic categorization that could be applied to any field of "purposive action" [9].

This study develops such a typology for unintended effects of international cooperation. The aim of this typology is to provide a scientifically robust framework to determine whether international development corporation "interventions have generated or can be expected to generate significant positive or negative, intended or unintended, higher-level effects" [10]. The proposed operational typology is based on a literature review that identified various potential types of unintended effects. To test the robustness of the typology, we applied it to all 644 evaluation reports from development cooperation projects involving the Netherlands between 2000 and 2020, using text mining and manual verification. This analysis contributed to a further refinement of the typology. The analysis of evaluations also underscored that the policy and evaluation community is in dire need of more academic guidance on comprehending and detecting unintended effects.

This paper is structured as follows: In Section 2, we first contextualize the study within the debate around unintended effects in the existing literature. In Section 3, we present the methodology that supported the two strands of the study. It first explains the methodology behind the operational typology and then looks at the text mining methodology. Section 4 presents the results. First, we present the typology, identifying 10 categories of unintended effects, and next we present the results applying and refining the typology to all publicly-available evaluation reports from the Dutch Ministry of Foreign Affairs using structured text mining. Finally, in Section 5, the discussion section, we highlight some of the limitations of this research and propose areas for further research.

## 2. Context

International aid has been criticized for almost as long as it has been around. Bauer already described in the 1970s how, for instance, development efforts could backfire if too much reliance on support through the governments was channelled, as often these governments did not have the interest of their population at heart [11]. This critical strand of literature has continued to attract followers such as Easterly and Nobel Laureate Deaton [12,13]. The latter is of the opinion that unintended effects of international aid on the quality of governance are so great that aid should be reduced as soon as possible: "Negative unintended consequences are pretty much guaranteed when we try . . . the pernicious effects are always there" [13]. Whereas especially earlier critics of foreign aid were rather general, more recently efforts have been made to better define and categorize unintended consequence of international aid [14]. How they are defined will be described in this context section and how they can be classified will be presented in Section 3 (operational typology).

The term "unintended consequences" refers to a particular effect of purposive action, which is different from what was wanted at the moment of carrying out the act and the want of which was the reason for carrying it out [9]. The word "unintended" is currently often interchangeably used incorrectly with "unanticipated", whilst they are not the same [15]. De Zwart shows the example of medicines; the doctor knows that prescribing a certain medicine can have unintended consequences (they are anticipated) but can still decide to go ahead regardless, because the intended effects outweigh the potential unintended consequences [15]. Hence, even though the doctor foresees the potential unwanted consequences, she presses ahead; the consequence was unintended but anticipated. Therefore, in our text, "unintended" effects refer to both foreseen (anticipated) and unforeseen (unanticipated) unintended effects.

Since the early 2000s, the Organization for Economic Cooperation and Development (OECD) has stipulated in its evaluation guidelines that evaluation of development pro-

grammes should deal with both intended and unintended results. This has been reaffirmed in the latest guidance of 2019, which states that evaluations focusing on impact ought to assess "The extent to which the intervention has generated or is expected to generate significant positive or negative, intended or unintended, higher-level effects" [10]. However, the little evidence available indicates that these guidelines are not followed systematically [16]. For example, a meta-evaluation of the United States Agency for International Development (USAID) monitoring and evaluation reports shows that only in 15% of the evaluations from 2009 to 2012 were unintended effects taken into consideration [17]. This percentage rose slightly from 2012 to 2015 according to de Alteriis [16]. An assessment of the evaluations of the Norwegian aid agency (NORAD) showed just 40% of Terms of References (ToRs) mentioned unintended effects. In one out of three NORAD evaluations for which ToRs requested evaluators to look at them, there was no mention of unintended effects, however. In general, unintended effects were dealt with in a superficial manner [7].

Attempts have been made to come up with classification schemes of unintended effects of international cooperation. The most used classification was developed by Jabeen and has been further elaborated by Koch and Schulpen and Koch and Kinsbergen [4,6,18]. In her categorization of unintended effects, Jabeen focused on four elements: their knowability, value, distribution and temporality [6]. Knowability refers to whether or not the unintended effects were anticipated. Value refers to whether the effects are positive, negative or neutral. Distribution refers to whether it is a spillover effect and whether it affects the non-target group or only the target group. Temporality refers to whether or not the effect happens simultaneously with the programme or after it. Koch and Schulpen added to this a fifth dimension, the "avoidability" of the unintended effect (could it have been mitigated?) [18]. In addition, a sixth type was encountered, the "fake unintended effect", in which people claimed an effect to be a surprise, but they planned for that unintended effect as part of a larger scheme. Lastly, Koch and Kinsbergen identified a final type, the "exaggerated unintended effect", in which opponents of a certain policy blow up potential negative side effects [4].

While these categorizations help to differentiate unintended effects once they are identified, they do not facilitate the identification of potential unintended effects in a systematic way. As a consequence, current overviews of unintended effects of international cooperation efforts in evaluations not only report a paucity of unintended effects (e.g., de Alteriis found 49 distinct unintended effects in 1369 monitoring reports and evaluations [16]) but also display a certain randomness and superficiality. Researchers often accidentally come across unintended effects in their evaluations and report these in a few lines towards the end of a report. They had neither planned to investigate them systematically nor had they developed appropriate research methods [7,16]. Intentionality about identifying unintended effects is necessary to address this. To become intentional, it is necessary to develop an operational typology that deals with the actual substance of the unintended effects.

## 3. Methodology

In what follows, we explain the methodological approach adopted in this study. Figure 1 provides a detailed diagram summarizing the two main strands within the study design. The first strand (3.1) develops an operational typology of unintended effects. It builds on and compiles the work performed in the past four years as part of an ongoing research programme on the unintended effects of international cooperation. The second strand (3.2) applies (and further refines) this typology to all evaluation reports from development cooperation projects involving the Netherlands between 2000 and 2020 using structured text mining.

### 3.1. Steps towards an Operational Typology of Unintended Effects

The new operational typology developed for this study serves as a starting point to facilitate better research into unintended effects of international cooperation, and it may evolve over time with new insights. International cooperation is defined here

broadly to include foreign aid but also other forms of international cooperation, such as peacebuilding missions. The typology is based on a review of different unintended effects that were identified in a four-year research programme called "the unintended effects of international cooperation" (2017–2021) (More information on the research program can be retrieved at https://www.ru.nl/anthropology/vm/unintendedeffects). This research programme comprises six structured literature reviews [4,18–22], 16 working papers (The special working paper series on this topic can be retrieved here: https://www.ru.nl/anthropology/vm/unintendedeffects), 13 peer-reviewed academic articles and an international academic conference [8]. Based on a review of this work, we identified 10 different types of unintended effects by applying a 3*3*3*3 rule, meaning that a type of unintended effect was included in our typology if it matched the following criteria:

It appeared in at least three different articles;
It occurred in at least three different domains of international cooperation;
It covered at least three different geographic areas; and
It was written by at least three different (groups of) authors (If a specific operational type of unintended effect has been observed in three different locations by different groups of authors, this does not mean that this unintended effect is bound to happen. Regularly, other researchers do not encounter these unintended effect in other contexts; yet the typology is still represented in this framework, as the aim of this typology is to support other researchers to analyse unintended effects in a holistic and systemic way).

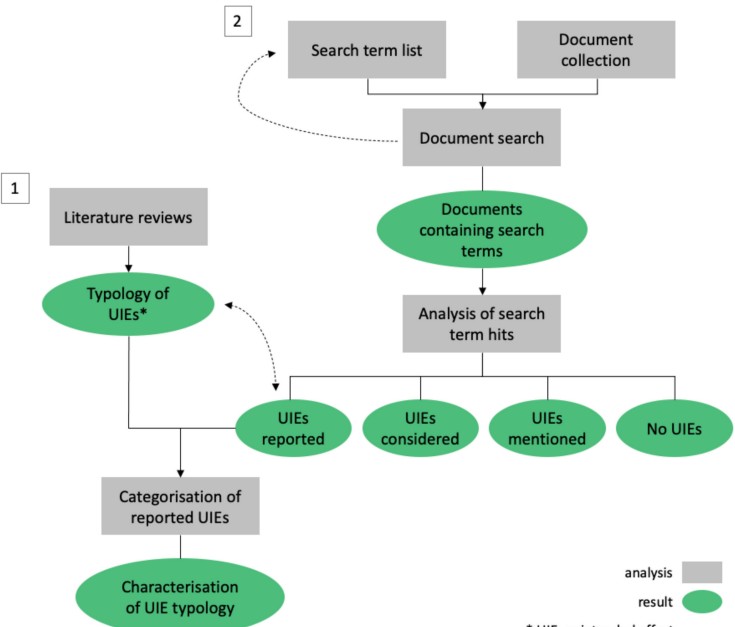

**Figure 1.** Flow diagram of the study methodology. * UIEs is short for unintended effects.

This process led to the identification of 10 different types of unintended effects. We introduce each of these in Section 4.

### 3.2. Text Mining Methodology

To test the typology, we next applied it to all publicly-available reports evaluating international cooperation efforts of the Dutch government from 2000 to 2020. The evaluations collected for the analysis were retrieved using web scraping, from the website (https://www.iob-evaluatie.nl/publicaties, accessed in 30 January 2021) of the Policy and Operations Evaluations Department (IOB) of the Dutch Ministry of Foreign Affairs. The website serves as a public repository that contains all public IOB evaluations and related documents, annual reports and ToR documents. Some older evaluation reports

were located at the IOB archive. The main reason for excluding publications before 2000 was that a relatively large share of these documents could not be made legible for the automated textual analysis. These evaluations were scanned images and therefore not readable as text by the text mining algorithm. Additionally, all publicly-accessible "decentralized evaluations" were retrieved; these are evaluations that cover Dutch foreign affairs policies but are not executed by the IOB but by the policy departments themselves (https://www.rijksoverheid.nl/ministeries/ministerie-van-buitenlandse-zaken/organisatie/beleidsevaluatie/decentrale-evaluaties-buitenlandse-zaken, accessed in 30 January 2021).

Altogether this includes 644 evaluations (excluding ToR documents and including case and country studies and synthesis reports), of which 356 reports were published in English and 288 in Dutch, after removing duplicates where a single evaluation existed in both English and Dutch. The majority (574 publications) were evaluation reports produced by the IOB, while 70 publications were decentralized evaluation studies. Appendix A provides an overview of the different sources of the documents.

A thesaurus of 359 search terms consisting of synonyms of "unintended effects" was created based on the researchers' expertise as well as on the existing literature on unintended effects. The full list with search terms can be found in Appendix B. Some search terms were formed by a combination of keywords (concordance searches), such as "unintended" and "effects", whereas other keywords were single-term entries, such as "externality" or "malpractice". The thesaurus was first created in English and translated to Dutch in order to search both Dutch and English documents. Because English terms are often used in Dutch documents (e.g., "trade-off" or "spillover"), the Dutch documents were searched with both the Dutch and the English search terms.

The textual searches were performed on each full-text document using Python, and a positive "hit" was recorded if (1) the keywords of the combined-term list co-occurred within the same sentence (the text is divided into sentences by splitting on punctuation defining a sentence, such as ".", "?" or "!"); or (2) a keyword from the single-term list was found in the document. For each "hit", the search results included the words found in the "hit" as well as the two sentences preceding and following the hit. Multiple hits could be returned for each document and were shown in separate rows of an Excel results sheet. The entire list of search term hits was subsequently analysed, primarily by going through the information generated by the automated searches. However, there was often not enough information to make an informed judgement based on these screen excerpts. In those cases, the full-text documents were reviewed to assess the nature of the findings and distinguish between the mere mention of unintended effects (false positives) and cases where actual unintended consequences were being reported. Finally, manual systematic screening removed the false positives and distinguished between three categories of documents resulting from the search. Table 1 provides a description of all four categories.

**Table 1.** Four categories of documents in which the search terms are found.

| Category | Explanation |
|---|---|
| "False positives" | Unintended effects are not mentioned; the hits did not return any relevant data. |
| "UIEs mentioned" | Unintended effects are mentioned, but in name only. In many cases, the explicit mention of unintended effects is part of the evaluation framework when authors refer to the OECD DAC guidelines or definition of impact. In some cases, reports contain an explicit statement that UIEs were left out of the evaluation, for different reasons (e.g., a lack of resources). |

**Table 1.** *Cont.*

| Category | Explanation |
| --- | --- |
| "UIEs considered" | The evaluators clearly announced a consideration of UIEs (and/or took steps to investigate them), but none were found or reported upon. |
| "UIEs reported upon" | UIEs were found and reported. These comprise both positive and negative unintended consequences, of any type. |

UIEs is short for unintended effects.

Our subsequent analyses zoomed in on the fourth category of documents—the reported upon unintended effects—to determine whether the effect had a positive or negative direction and how the effects could be categorized in terms of the 10 unintended effects of the typology. The results of the automated textual search complemented by the manual verification process give insight into what unintended effects occur in international cooperation programs.

Section 4 presents the results—first the operational typology, followed by the results from the text mining.

## 4. Results

### 4.1. Ten Types of Unintended Effects: An Operational Typology of Unintended Effects of International Cooperation

The review and compilation process within the first strand of this study led to the identification of 10 different types of unintended effects, which we clustered in four "levels": micro-, meso-, macro- and multilevel. Figure 2 provides the overview, followed by an explanation of each. Six of the 10 types were classified as appearing on a single level, while four were classified as being across multiple levels. While this distinction is not always clear-cut in practice, we argue that it is conceptually (and visual) useful.

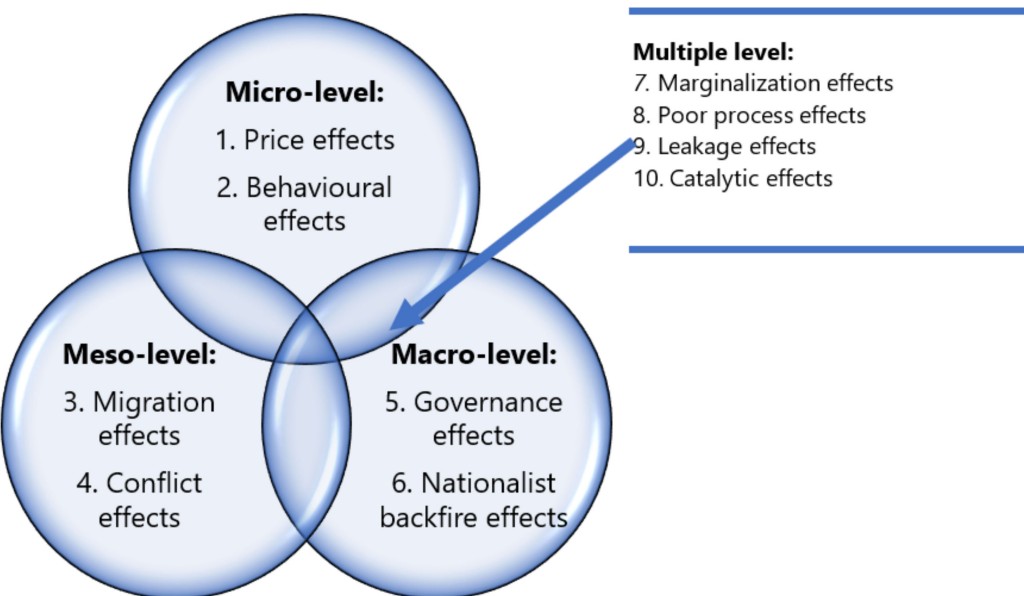

**Figure 2.** Operational typology of unintended effects of international cooperation.

4.1.1. Micro-Level: Price and Behavioural Effects

The micro-level unintended effects that appear in the literature are those unintended effects that are predominantly positioned at the individual and household level. It includes two types: unintended price and behavioural effects.

Price Effects

Unintended price effects occur when the external intervention distorts local prices—for example, by strongly pushing or depressing demand and/or by creating monopolies/monopsonies, or by creating certification costs. Sectors in which price effects have been substantial include food aid, (conditional) cash transfers and responsible trade initiatives.

When it comes to food aid, the arrival of external food supplies has had the intended effect of depressing the price of food in the target area. However, as an unintended effect, it can also reduce the incentive for local farmers to produce food, leading to lower food production in the long run, as evidence from Zimbabwe and Ethiopia has shown [23–25]. This unintended effect has been known for some decades, and the most prominent organization in the field of food aid, the World Food Program, has become aware of this and tries to mitigate the effect by purchasing more food locally [26].

Cash transfers (conditional or unconditional) are a relatively new form of international cooperation, and studies highlighting their unintended price effects are thus recent publications [27–29]. One such effect is constituted by a rise in food crop prices due to a large influx of cash in a community, which can ultimately lead to increased malnutrition among non-beneficiaries. Research in the Philippines showed for instance that stunting rates among young non-beneficiary children increased by 11 percentage points, with even greater increases in the most saturated areas [29].

A last sector in which price effects play a role are various responsible trade initiatives in which, by means of ethical standards, farmers and miners are supposed to benefit more from international trade. While there is ample evidence that traders and miners benefit, there is a group of studies now indicating that there are also costs related to these trading schemes [30–32], either directly (because of certification costs) or indirectly (because of a monopsonic market system in which the ethical trade labels become the price setters) impacting communities.

Behavioural Effects

Unintended behavioural effects occur when the psychological reaction of the recipient or affected persons of the external intervention differs from the initial prediction. Behavioural effects can take various forms, such as the rebound effect (in which for instance energy savings are offset by more use by consumers) or motivational effects (in which the intrinsic motivation is reduced because of an external intervention). The three domains in which behavioural effects have been encountered systematically are payment for environmental services, sexual and reproductive health and rights and (gender relations within) microfinance programmes.

Payment for environmental services is a rather new form of international cooperation in which providers of environmental services (e.g., maintaining forests) are financially compensated for providing those services. Research shows that unintended effects can occur if the intrinsic motivation of people to protect the forest is reduced [33–35]. Hence, the unintended effect that may occur is that the risk of deforestation becomes higher after the end of such a programme when payments stop. For example, a study in Mexico found that receiving payment for environmental services tended to make the future of conservation contingent on monetary and utilitarian reasons [33]. This effect is also referred to as the motivational crowding out effect.

In the field of sexual and reproductive health and rights support programmes, the behaviour of intended beneficiaries regularly turns out differently than expected [36–38]. Research in Uganda [36] found for instance that a programme aimed at strengthening safe sex practices actually led to an increase in male promiscuity. The behavioural response of men diverged from the expectation because of the wrong assumption that men would simply accede to the increased female demand to use contraceptives. A common denominator in this gap between expectations and outcomes appears to be that programmes are developed in one setting or for one age group and then copied to other settings thereafter.

While the intervention is suitable for the setting it was developed for, it might not be for the settings it is copied to, leading to unintended behavioural effects [37].

These unintended behavioural effects have also been documented in the field of microfinance, especially with respect to gender relations [39–41]. Many microfinance programmes focus mostly or only on women. Research has shown that increased financial power of women in the household can have behavioural responses by spouses, including increased domestic (verbal and/or physical) abuse by male partners [41].

### 4.1.2. Meso-Level: Conflict and Migration Effects

Meso-level unintended effects are those effects that are primarily active at the level between the micro- and the macro-levels, and this is the level, for instance, of the community, the organization or the town, or the relations between communities and organizations. For the meso-level, prior research has found conflict and migration unintended effects.

### Conflict Effects

An external intervention can sometimes engender tensions between communities, for instance a perceived unequal treatment of (recipient and non-recipient) groups and communities. These unintended conflict effects have been documented in development cooperation areas as diverse as nature conservation and post-natural disaster humanitarian aid, especially in already conflict-affected and high-risk areas. The conflict-sensitive programming that has surged over the last decade in the humanitarian and development sector can be seen as an acknowledgement of the conflict effects of aid and has aimed to mitigate these side effects [42]. Mitigation of the conflict effects has received the most attention out of the different unintended effects over recent decades.

The literature review showed that in the domain of nature conservation, conflict effects are particularly prevalent [43–45]. This appears related to the demarcation of certain areas as protected (or not), which can lead to the re-opening of frozen conflict, as communities might need to be displaced, access to gathering and hunting grounds are restricted and benefits (of for instance tourism) need to be shared.

Additionally, in the domain of post-natural disaster humanitarian aid, unintended conflict effects are widely documented [46–48]. What these studies observe is that in an emergency setting, external donors are often challenged by a lack of contextual information and can thus overlook local tensions. By working with intermediaries or providing support to certain groups (e.g., along ethnic lines), these local pre-existing tensions can be aggravated.

Especially in areas that are already fragile, aid risks catalysing conflict [49–51]. In countries such as (South) Sudan and Syria, humanitarian aid has been documented to generate extra conflict, either directly or indirectly. It leads directly to conflict if actual fighting erupts to gain access to depots of humanitarian aid, or indirectly if it allows rebels to regroup and re-energize, as was the case with the Hutu genocidaires in the refugee camps around Goma in 1996 [52].

### Migration Effects

External interventions can create extra economic and security pull and push factors, resulting in populations moving towards or from an area of intervention. In development research, migration effects have often been overlooked [53], partly because migrants are often forgotten in pre- and post-test effect measurements, as they are either absent in the pre-test (inward migration) or the post-test (outward migration). Migration effects have been found in various thematic areas, such as peacekeeping, social and humanitarian programming and (conditional) cash transfer. Since migration decisions are ultimately individual decisions, these unintended effects could also be described as individual-level effects, but since inward and outward migration can affect the composition of entire cities, we treat them here as meso-level effects.

The migratory effects of peacekeeping missions are well documented [54–56]. When internally displaced people flock to the area around a blue-helmet compound, the effects can be short-term. They can also be long-term, as a study of urban planning in Goma, DR Congo, showed, where the long-term presence of peacekeepers attracted international NGOs and their staff [54]. The large influx of expatriates led to an increase in house prices and gentrification.

In addition, inward migratory effects have also been documented because of humanitarian and social programmes, in countries such as Laos, Malawi and Syria [57–59]. The provision of health services in Malawi by external actors has operated as a pull factor for the local population, who decided to move closer to these areas where adequate health care was provided by external actors.

These findings are corroborated with research on the migratory effects of cash transfers [60–62]. Cash transfers have been found to both stimulate inward migration and act as a break on outward migration, whether these payments were linked to provision of ecosystems or more general conditional cash transfers [60,61]. If the cash transfers are, however, compensation payments for the destruction of livelihoods (e.g., the construction of a dam), these payments are linked to outward migration [63].

### 4.1.3. Macro-Level: Governance and Nationalist Backfire Effects

Macro-level effects are those unintended effects that are principally situated at the national level. In the context of the development of the operational typology, "governance effects" and "nationalist backfire" effects emerged.

### Governance Effects

External interventions can have a positive or negative indirect impact on the quality of institutions at any level at the recipient country because of changing accountability relations and relaxed budget constraints. The governance effects of foreign aid have been a hotly contested academic topic for decades [64], and opponents of international aid have claimed that the pernicious governance effects of aid are always there and that therefore international aid needs to be abolished altogether [13]. Unintended governance effects have been found to be contingent on initial levels of governance; if the governance level was already appropriate, the intervention could help to strengthen it, but if the quality of governance was already worrisome, incoming governmental aid could further exacerbate the quality of governance [65]. Donors tend to adjust their aid modalities to reduce negative governance effects (e.g., providing only budget support for countries with good levels of governance) [66]. However, sometimes geopolitics gets in the way of these side effect mitigation measures; in that, case aid is transferred to recipient governments despite indications that this might contribute to, for instance, increased authoritarian tendencies [67]. Unintended governance effects have been encountered with respect to budget support, debt relief and also as a result of migration management programmes.

Budget support is the aid modality in which donor governments transfer funding to the general account of the recipient government, sometimes with certain governance conditionalities. Of the various forms of aid modalities that exist, this form of external support is most associated with unintended governance effects, as it provides the recipient government with fully fungible funds [68–70]. Even though the evidence is mixed, budget support has also been found to contribute to a reduction in the social contract between the population and the government.

Debt relief has been found to have similar unintended governance effects as budget support, as it also relaxes the budget constraint a government faces. Debt relief is not just linked to a change in general governance levels, but especially to corruption [71,72] and also to repression levels [73].

While the debates on the governance effects of budget support and debt relief have been ongoing for quite some time, the governance effects of migration management programmes have only recently surfaced within broader debates [74–76]. This can be explained

by the relative relentlessness of these programmes. While this debate is fresher, similar patterns can still be detected in which inward flows of financing to governments with poor levels of governance are said to contribute to more repression and even less accountability towards the local population.

Nationalist Backfire Effects

An unintended nationalist backfire effect occurs when a domestic political actor that risks seeing its relative power decline succeeds in bolstering its position or policy by casting the development cooperation as foreign intrusion. Nationalist backfire effects have been observed in the international cooperation fields of democracy promotion, LGBTIQ+ rights promotion campaigns and gender programming. Whereas protocols have been developed by donors to enhance conflict sensitivity or reduce the risk that aid is embezzled, no safeguards have been enacted with respect to nationalist backfire effects.

Democracy promotion has been one of the cornerstones of the policies of some of the major international donors such as USAID. While the intended objective of these interventions is strengthening democratic forces in societies, occasionally the opposite has happened [77–79]. Some autocratic governments have won substantial public sympathy by arguing that opposition to Western democracy promotion is resistance not to democracy itself but to American interventionism [77]. Pikulik and Bedford (2019) describe why democracy promotion programmes are maintained in spite of clear negative unintended effects.

Some of the findings with respect to LGBTIQ+ rights promotion campaigns are similar to those with respect to democracy promotion programmes [80–82]. The president of Uganda, Yoweri Kaguta Museveni—facing a popularity crisis—was successful in bolstering his own position and clamping down on gay rights by claiming that homosexuality was something "un-African" [81], to be resisted as a new form of colonialism. In the end, he was successful in strengthening anti-gay laws and increasing the related prosecution of LGBTIQ+ individuals, which was the opposite of the intended objective.

Additionally, gender promotion programmes have been found to occasionally suffer from similar nationalistic backfire effects [83–85]. This was especially the case when gender programming was concomitant to a foreign military intervention, as was the case in Afghanistan. Those who stand to lose from the foreign intervention, in this case the Taliban, cast their opposition against more rights for women and girls in terms of fighting a hostile occupying force [84].

4.1.4. Multiple-Level Effects: Marginalization, Leakage, Catalytic Spillover and Poor-Process Effects

Some of the unintended effects above can operate at more than one level. For example, as indicated, migration effects are not exclusively meso-level effects, and price effects could, in theory, be national if a development cooperation intervention were sufficiently large. In the literature, four multiple-level unintended effects have been identified: marginalization, poor-process, leakage and catalytic effects.

Marginalization Effects

Marginalization unintended effects arise when a development cooperation-supported intervention does not succeed in reaching (or may actually be hurting) those who are already relatively marginalized, setting in motion a process of increasing inequality. Marginalization effects can manifest themselves regardless of whether the intended objective is achieved (e.g., biodiversity is restored). Nature conservation efforts and infrastructure development in multiple settings have been associated with unintended marginalization. At the macro-level, international aid has been found to be strengthening the position of local elites, contributing to the relative marginalization of others.

Well intentioned nature conservation programmes do occasionally contribute to the deprivation of the local population whose livelihoods are dependent on that natural resource, as evidence indicates from Mexico and South Africa, amongst other countries [86–88].

Through enclosure, the local population living close to the—now better "protected"—national parks see their access to their livelihoods restricted (this effect is closely intertwined with the conflict effect mentioned in Section 3.2). The protection of these parks is sometimes enforced by means of "green violence" against the local population, further marginalizing such individuals [89]. While there are social and environmental safeguards of, for instance, forestry programmes (e.g., the Redd+ safeguards [90]), these are not always upheld [22].

Additionally, mega infrastructure projects, such as dams, highways and even wind parks have been associated with marginalization of the local population, whenever population members have been forcibly resettled without free prior and informed consent. Often, such forced relocation is associated with a net reduction of access to livelihoods [91–93]. While safeguards that ought to ensure free prior and informed consent are now part and parcel of the due diligence procedures of large (publicly backed) international investors in these programmes, research indicates that the implementation difficulties are widespread [94].

Lastly, the distributional impact of foreign aid inflows can also be skewed to the local elites, leading to a sense of relative marginalization [95–97]. If a part of the aid is siphoned off when it enters the country, or when it is directed to sectors or regions that are of particular interest to the elites (e.g., private health care or education), this might increase inequality within a society.

Leakage Effects

Leakage effects, also known as displacement or waterbed effects, materialize when communities and areas outside of the project/programme intervention area are affected because of the intervention. Leakage effects can exist in nearly all domains of international cooperation whenever the international funding attracts talent to the intervention area, away from the non-intervention area (e.g., teachers or judges). This capacity then "leaks" to the intervention area. However, the leakages can also go in the other direction; for example, criminality is not reduced but simply moves to a non-intervention area. Leakage effects have been documented in the field of forest conservation, public sector development and health systems.

In forest conservation there is a risk that loggers might just move to a less protected area. While this effect is well known in the academic literature [98–100], this is not systematically addressed in the evaluations of forest conservation programmes themselves [22].

Leakage effects are also well documented in the public sector domain. The brain drain from local and national governments to international actors has been especially well researched [101–103]. While many international actors aim to strengthen the capacities of local and national governments, they keep the best performing civil servants for their own agencies. The unintended effect is that the capacity of those governments may leak away.

Similar types of leakage effects have been observed in the health domain. To combat the AIDS pandemic, amongst other health crises, international actors have been found to choose to set up "vertical" health programmes. While these programmes are quite effective in reining in various serious health threats, they have also contributed to a weakening of the overall health system [104–106]. The current emphasis on "health systems strengthening" can be seen as a reaction to the negative side effects of such vertical health programmes [107].

Catalytic Spillover Effects

Catalytic effects are positive spillover effects from the intervention area, beneficiary or domain to the non-intervention area, non-beneficiaries or non-intervention domains. They can be considered unintended if the donor or implementing agency did not intend such positive spillovers to happen. These catalytic effects have been documented in such domains as agricultural innovation, women's rights and health and nutrition programming.

For instance, in areas where agricultural training is provided and improved seeds are distributed, the impact of the intervention is often analysed by measuring productivity within the intervention area. However, this way of measuring might underestimate the

total scope of the effect, as farmers outside of the intervention area may start to copy the behaviour of the more productive farmers within the intervention area. Such positive spillover effects have been documented in countries such as Ethiopia [108], Kenya [109] and Peru [110].

In this study, some negative effects with respect to women's rights were found, such as the negative behavioural effects of spouses towards their financially empowered wives (Section Behavioural Effects) and the nationalist backfire effects with respect to gender programmes (Section Nationalist Backfire Effects). There are, however, also catalytic effects related to women's rights programmes. Whereas the intended objective is to improve women's rights, the improvement also has a positive effect on other domains such as reduced rates of infant mortality [111–113].

Lastly, nutrition and health programmes also have had a catalytic effect on school performance rates. Whether it relates to the well-known deworming initiatives or early childhood nutrition programmes, the positive effects of these interventions on test scores in schools are well documented [3,114,115]. In some cases, the catalytic effects start off as being unintended, but as the evidence pours in, agencies often include the catalytic effects in their theories of change by altering them from unintended into intended effects.

Poor-Process Effects

Compared to all the other unintended effect inventories in this research, poor-process effects are situated at a different level, as they hint more at the origin of the unintended effect than at the substance of the effect itself. These effects were nevertheless included here, as otherwise the unintended effects that fall in this category risked being easily overlooked. They constitute a particularly relevant subset, as these unintended effects fall within the sphere of control of implementing partners, for example. These effects could be overcome when the (aid) process is implemented better. Poor-process effects arise because of poor processes of implementation and coordination of external interventions, which contribute to avoidable moral hazard (e.g., in the domain of peacekeeping), avoidable stigmatization (with respect to for instance conditional cash transfers) and unnecessary hype (in the field of sexual violence for instance).

If peacekeeping operations are not executed with a zero-tolerance policy with respect to sexual abuse and exploitation, there is a high risk that a significant proportion of the peacekeepers will engage in this type of behaviour [116–118]. This has been documented in areas of conflict as diverse as Kosovo, Liberia and Afghanistan. Because the United Nations is now taking a more active stance against exploitative behaviour of peacekeepers, the number of allegations of sexual exploitation and abuse committed by personnel serving with the United Nations (which includes peacekeepers) is declining [119].

Stigmatization is another avoidable unintended effect, which is also the result of poor processes. Whereas stigmatization can be found in various parts of the international aid system (e.g., in the field of care that is given to survivors of sexual violence), it has been documented systematically in the field of conditional cash transfers [120–122]. While living in poverty is often associated with a stigma, receiving (international) aid can also comprise stigmatization if the support is, for instance, provided in an indiscrete way.

Additionally, at the systemic level of international cooperation, poor processes can generate unintended effects. What sometimes happens in the international aid architecture is that a hype emerges, which can lead to unnecessary and negative side effects. These side effects can be avoided by developing better cooperation relations among the different aid actors. This has been researched in depth in the field of the response to sexual violence in the DRC [123–125]. Whereas sexual violence in the DRC is a real problem, the excessive number of aid actors it attracted and the hyped, narrow attention given to one particular injustice at the expense of wider conflict dynamics and the underlying root causes of gender-based violence have led to substantial side effects, such as an increase in fake testimony about rape and the inappropriate sentencing of innocent civilians [124].

### 4.2. Applying the Typology to Evaluations of Dutch Development Cooperation

The results of the automated textual search are summarized in Table 2. The documents that contained no hits are included as category 0—no search terms. Unintended effects were found in 446 evaluation documents (69.3% of the evaluations), with the remainder 198 publications in which no search terms were found. However, more than half of these 446 documents—287 documents altogether (44.6% of the total)—returned false positives upon manual verification. In 58 documents (9.0% of the total), they were mentioned, but in name only. In approximately one in six evaluation documents (15.6%), unintended effects were either explicitly considered but none found (1.2%) or considered and reported upon (14.4%).

**Table 2.** Categorization of documents in the data set.

| Category | Number of Publications (Share) |
| --- | --- |
| No search hits | 198 (30.7%) |
| "False positives" | 287 (44.6%) |
| "UIE mentioned" | 58 (9.0%) |
| "UIE considered" | 8 (1.2%) |
| "UIE reported upon" | 93 (14.4%) |
| Total | 644 (100%) |

Figure 3 provides a breakdown of the unintended effects across years. The category "Yes" in Figure 3 refers to a document reporting on unintended effects (category 4 in Table 2), while "Considered" refers to category 3. "No" includes all other documents. The reporting of unintended effects is neither systematically increasing nor decreasing over time. Appendix C provides a further breakdown showing the types of documents reporting upon unintended effects. Most documents are in the category "evaluation", but a relatively high number of documents are in the category "external evaluation", which shows that external evaluators also look into unintended effects ("external" refers to those evaluations that were executed by other departments of the Ministry of Foreign Affairs). For the external evaluations, in 14 documents unintended effects were reported upon, of the total of 70 external evaluations. This means that in the external evaluations, unintended effects were reported upon in 20% of the documents, which is a slightly larger share compared to the overall percentage of documents where unintended effects were reported upon (14.4%, category 4 in Table 2).

#### 4.2.1. Types of Unintended Effects

Altogether, we identified 177 individual unintended effects across the evaluations that reported upon them (93 evaluations). We next applied the proposed typology (introduced earlier in this article), categorizing every unintended effect according to its direction (either "positive" or "negative"), and according to the 10 different types of unintended effects.

The typology was designed to be operationally relevant. A first test was therefore whether the application of the typology is practical and meaningful to the types of unintended effects that are found in (Dutch) development cooperation interventions, which we found to be the case. When manually applying the typology to these 177 unintended effects, the analysis found 9 of the 10 unintended effects categories to be represented (no unintended effects in the category nationalist backfire effects were found). In only five cases (2.8%) were unintended effects found that fell outside our proposed typology. These findings suggest that the typology is neither overly broad (in which case various categories of unintended effects under the typology would not have been represented among the actual ones found) or overly narrow (in which case there would have been a high proportion of actual unintended effects found that could not be classified).

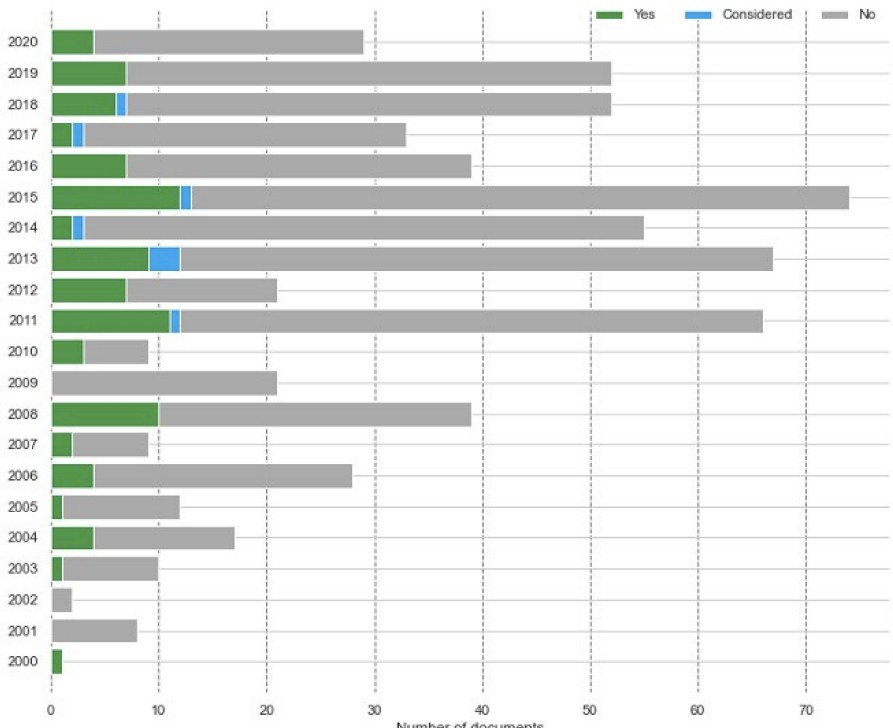

**Figure 3.** Annual number of evaluation documents, broken down by inclusion of unintended effects.

The breakdown of unintended effects per category, sorted by "negative" or "positive", is shown in Figure 4. First, in contrast perhaps to a public perception driven by newspaper headlines, the figure shows that the unintended effects found in the evaluations were relatively balanced across both positive (44%) and negative (56%) effects. Second, we found that most unintended effects were found in the multiple-level cluster, with the (positive) catalytic spillover effect category representing the highest number followed by the (negative) poor-process effect category. Unintended governance effects (both positive and negative) and unintended behavioural effects (most negative) were also relatively common, as were negative leakage effects.

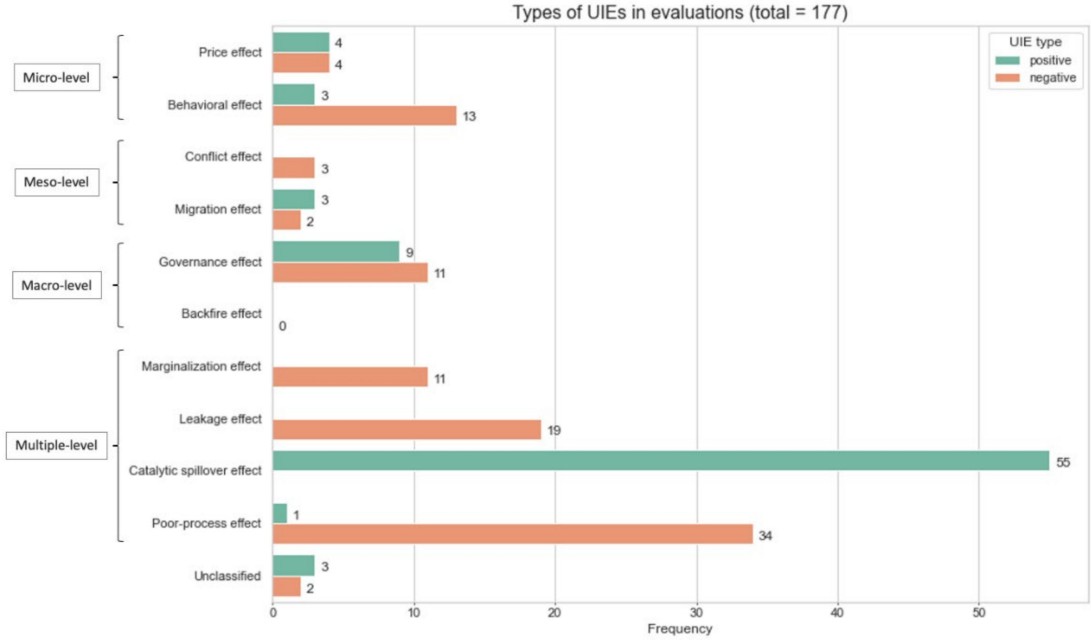

**Figure 4.** Categories of unintended effects in evaluations.

### 4.2.2. Descriptions of Encountered Unintended Effects per Category

Micro-Level Effects

The price effects were encountered in a variety of thematic areas, such as food security and sustainable commodity production. The evaluation "Riding the Wave of Sustainable Commodity Sourcing—Review of the Sustainable Trade Initiative IDH 2008-2013" noted a negative price effect of the sustainable trade programme for local farmers [126]. Sustainable trade initiatives support market transitions in which social and ecological certification plays an important role. However, it was found that it "constrain[s] the competitive environment for the farmers, reducing their bargaining power and making them more dependent on one particular outlet. By supporting a single trader's efforts—even when this company has won an open call for proposals—the public funding intervenes in the market structure and may restrict competition".

Behavioural effects were also found in diverse sectors, such as access to clean energy (e.g., cook stoves) and water. The evaluation "The Risk of Vanishing Effects—Impact Evaluation of Drinking Water Supply and Sanitation Programmes in Rural Benin" highlighted how households changed their behaviour once they got access to clean water: "An interesting but worrying negative unintended effect of the installation of improved water points is that households discontinue point-of-use water treatment to improve water quality" [127].

Meso-Level Effects

Migration effects occurred in a variety of sectors and were labelled to have both a positive and a negative direction. A positive unintended migration effect was mentioned in an evaluation of the ORET programme, an abbreviation for Development Relevant Export Transactions. The evaluation, entitled "Work in Progress—Evaluation of the ORET Programme: Investing in Public Infrastructure in Developing Countries", focused on the trainings (in the Netherlands) that were provided to engineers from low- and middle income countries that participated in the trainings that accompanied the supported export transactions: "The country benefits indirectly from the remittances of the engineers who have moved abroad and who are expected to return with more experience" [128].

The conflict effects were negative and were found also in a diversity of thematic areas and regions. The consequences of the lack of an adequate implementation of the "do-no-harm" principle were especially highlighted, for instance in the evaluation "Aiding the Peace: A Multi-Donor Evaluation of Support to Conflict Prevention and Peacebuilding Activities in South Sudan 2005–2010" [129]. The evaluation noted that "unnecessary tensions were introduced by the failure to incorporate a 'do no harm' approach. One example was the rebuilding of a damaged water dike in one village that negatively affected villages downstream. Another was the placing of a health clinic intended for mutual use in only one of two villages in conflict with each other", indicating that the activities in fact fuelled conflicts in the region.

Macro-Level Effects

The governance effects were both positive and negative. The "Evaluation of the Dutch Food Security Programme in Ethiopia", which included an impact study of the Integrated Seed Sector Development Project (ISSD II), found unintended positive governance effects, as the government scaled the food security governance experience in one region up to the other regions: "the federal government used the experience of the project to scale-up land use planning in other regions. Moreover, the federal government is using the project experience as basis for developing the national land-use policy" [130].

The academic literature has provided evidence of backlash effects in a variety of domains, such as gender and democracy promotion (Section Nationalist Backfire Effects). Even though the Dutch international cooperation programme was very active in these domains in the period 2000–2020, and numerous evaluations were carried out on these themes, this unintended effect was not reported.

Multiple-Level Effects

In sectors such as private sector development and food security policies, the marginalization effect (always negative) surfaced. For instance, the "Mid-term Evaluation of the Netherlands Food Security Programme in the Palestinian Territories" noted the side effects of the promotion of high-value production (HVP) crops [131]. "The required capital investments HVC production is rarely accessible to small farmers and that further strengthens the better off farmers and increases their power to compete over natural resources with their poorer colleagues".

Evaluations registered leakage effects (by definition negative) in various domains. For instance, the evaluation "Drinking water supply and sanitation programme supported by the Netherlands in Fayoum Governorate, Arab Republic of Egypt, 1990–2009" focused on a side effect of the increased water availability [132]. "An unintended effect of the extension of the water network is that improved water availability has in some locations contributed to the overflowing of on-site sanitation tanks and possible health dangers". This is considered a leakage effect as the external intervention had an effect in a different sector (health) than the intended sector (water).

Catalytic spillover effects (by definition positive) were also found in many different domains, ranging from health programming to private sector development. One evaluation, called "Work in Progress—Evaluation of the ORET Programme: Investing in Public Infrastructure in Developing Countries", showed how work on a port (focused on improving transport) had an impact on a different sector (tourism): "An unintended consequence of the ORET-financed words is the unexpected increase in tourism due to the extended beach" [128]. This is a catalytic spillover effect, as the unintended effects were found in a different sector (tourism) than the targeted sector (transport).

Lastly, poor-process effects also came forth in a wide array of different evaluations and sectors. One evaluation, "Humanitarian and Reconstruction Assistance to Afghanistan, 2001–2005", aimed to shed light on the consequences of external intervention on interpersonal and inter-organizational relations in Afghanistan [133]. It argued that "Hierarchical relationships between the international project national staff and the civil servants, and competition between international funded projects, have been unintended consequences".

## 5. Discussion

### 5.1. Main Findings

Major international development agencies represented in the OECD DAC have been stipulating since the early 2000s that evaluations of development programmes should include unintended results. However, the limited evidence systematically documenting this suggests that this is still rarely happening. A common challenge to its inclusion in evaluations has been the lack of a common framework—a typology—around unintended effects that is both rigorous and operationally meaningful.

Building on a four-year research programme, we identified 10 types of unintended effects that have been commonly found in the literature on international development cooperation and grouped these into four clusters: micro-, meso-, macro- and multi-level. Using automated text searches and manual verification, we applied this typology to all publicly available evaluations for the period 2000–2020 of one of the largest international aid agencies: the Dutch government's development cooperation programme.

Consistent with other findings on unintended effects reported upon in evaluations of USAID and NORAD, we find that most evaluations of Dutch development cooperation (interventions) do not report on unintended effects. Only about one in six evaluations (15.6%) meaningfully consider unintended effects; this is 101 out of 648 evaluation reports. Of these, 93 evaluation reports find a total of 177 unintended effects. When applying the typology, all but 5 of these 177 unintended effects can be assigned to an effect in our operational typology, and all but 1 of the 10 types within the typology have indeed surfaced in our analysis of evaluation reports (several times), indicating that the proposed categories

are neither too broad nor too narrow. Furthermore, in contrast perhaps to public perception, these unintended effects are fairly balanced between positive and negative outcomes.

### 5.2. Limitations

Naturally, there are several other limitations to this study. For example, we have followed the evaluations in their assessment of unintended effects. All effects that were labelled or written down as an unintended effect were hence included in this study, and the findings of evaluations were not categorized on, for instance, societal significance or reliability of the reporting on the unintended effects. Furthermore, the proposed typology is not fully consistent with the *mutually exclusive, collectively exhaustive* (MECE) principle. Some unintended effects could fall into more than one category, and the fact that we could not classify all unintended effects indicates that the categories are not exhaustive. Perhaps when applied to evaluations by other development cooperation programmes, other types of unintended effects may emerge. However, because the typology of types of effects is based on a broad-based review of the literature—not limited to Dutch development cooperation—we expect that this typology will similarly apply elsewhere.

### 5.3. Relevance for Policymakers and Evaluators

The intent of the proposed typology is to support policymakers and evaluators become more intentional about unintended effects by providing both rigorous (based on academic assessments of unintended effects) and operationally relevant guidance. This should trigger evaluators to think critically about possible unintended effects and to consider these in the design of new programmes, their evaluations and subsequent programme modifications, if any. Similarly, at the policy level, more careful consideration of unintended effects can guide the development of new policies and the direction of existing ones.

To ensure that the typology indeed supports more careful consideration of unintended effects, it will need to be *operationalized* in relevant workflows. For example, in the development of theories of change and/or logic models for interventions, reflection of unintended effects should be required. Similarly, ToR documents for evaluations need to explicitly call on evaluators to assess unintended effects.

We have examined whether or not ToRs urge evaluators to monitor and evaluate for unintended effects, and how specific the instructions are (what to look for). The results can be found in Appendix D. Two-thirds of the 90 ToR documents do not include the study of unintended effects in the planning or evaluation framework, even if ex post evaluators suggest that they might have occurred, as was mentioned in one evaluation, for example: "The negative spill-over effects of the private standards programmes did not receive much attention. Examples of possible negative spill-over effects include more rainforests being destroyed for (less remunerative) crops instead of palm oil and a shift in child labour from cocoa plantations to even more miserable jobs. These effects also fall outside the scope of this evaluation, as they do not affect trade" (Better Ways of Trading—Evaluation of Technical Assistance for Trade Policy and Regulations (2017), page 96). Only four ToRs make explicit reference to the standard OECD framework question on unintended effects (e.g., "What are unanticipated positive or negative consequences of the programme?") and additionally highlight examples of possible unintended effects that could occur and should be taken into account.

Furthermore, beyond requiring reflection on unintended effects at the (re-)design and evaluation phases, agencies may also need to address more subtle implementation barriers. For example, several evaluations mentioned that unintended effects were not included because the methodology or scope of the evaluation was not sufficient or did not allow for the examination of unintended effects. For example, one evaluation mentioned that "In particular with regard to the indicator on 'job creation' it is recommended to broaden the view to 'income generation' and to ensure that not only the direct but also indirect impacts and co-benefits are taken into account" (Strategic Evaluative Review of the Energising Development Partnership Programme, short version (2018), page 21). A

different evaluation says this: "Also, the surveys are not designed with the attempt to measure whether the process actually leads to increased effectiveness and whether there are unintended effects of the processes of change set in motion" (Framework Terms of Reference for the First Phase Evaluation of the Implementation of the Paris Declaration (2007), page 3). These quotes suggest that while evaluators are aware of the fact that unintended effects may occur, they do not always have the means or right approach to capture such effects. These examples merit a closer look at what the specific barriers are that inhibit evaluators' abilities to integrate the study of unintended effects within evaluation studies and what may adequately remedy such challenges.

In sum, the operational typology developed in this research is an important step toward more intentional consideration of unintended effects. To fully operationalize this in practice requires integration into programme design and assessment workflows, with careful attention to more subtle implementation barriers.

**Author Contributions:** Conceptualization, D.-J.K., J.d.L.; writing—original draft preparation, D.-J.K.; methodology, J.V.; data curation, J.V.; validation M.v.d.H.; formal analysis, E.T.; funding acquisition, J.d.L.; supervision J.d.L.; writing—review and editing, D.-J.K., J.V., M.v.d.H., E.T. and J.d.L. All authors have read and agreed to the published version of the manuscript.

**Funding:** This research received no external funding.

**Institutional Review Board Statement:** Not applicable.

**Informed Consent Statement:** Not applicable.

**Data Availability Statement:** The data presented in this study are available in Appendix A Table A1.

**Acknowledgments:** The authors would like to thank the sounding board of this study for useful comments on the research throughout the process. The sounding board consisted of representatives of both the OECD, The Netherlands Ministry of Foreign Affairs and the independent Operations and Evaluations Department of the Netherlands Ministry of Foreign Affairs. In addition, valuable suggestions were received during a roundtable discussion with this Operations and Evaluations Department during a roundtable on 1 July 2021. All remaining errors are the responsibility of the authors.

**Conflicts of Interest:** The authors declare no conflict of interest. The organization for which Dirk-Jan Koch works, the Ministry of Foreign Affairs, has neither interfered with the design of the research nor with the outcomes.

## Appendix A. Sources of Documents

**Table A1.** Sources of the evaluation documents.

| Type Document | Downloaded from | Missing Documents Due to Errors * | Deleted Documents | Final Number of Documents |
|---|---|---|---|---|
| IOB evaluations | https://www.iob-evaluatie.nl/publicaties | 4 | | 405 |
| Decentralised evaluations | NL: https://www.rijksoverheid.nl/ministeries/ministerie-van-buitenlandse-zaken/organisatie/beleidsevalu-atie/decentrale-evaluaties-buitenlandse-zaken EN: https://www.government.nl/ministries/ministry-of-foreign-affairs/organisation-al-structure/ministry-of-for-eign-affairs-evaluations/decentral-evaluations-foreign-affairs | 14 | | 76 |
| Evaluations IOB archive | http://archief.iob-evaluatie.nl/publicaties.html | 18 | 72 ** | 589 |

**Table A1.** *Cont.*

| Type Document | Downloaded from | Missing Documents Due to Errors * | Deleted Documents | Final Number of Documents |
|---|---|---|---|---|
| | Total | | | 1071 |
| | Deleted document types | | 22 *** | |
| | Deleted documents before 2000 | | 19 | |
| | Duplicated documents (IOB archive and IOB) | | 120 + 116 | 236 |
| | Total | | | 794 |
| | Documents not in English or Dutch | | 60 | |
| | Terms of Reference documents | | 90 | |
| | Total | | | 644 |

* Examples are documents that contained only tables, meaning the text cannot be extracted, or documents with a link that led to an "Error 404: Page not found". ** Scanned documents which are an image and cannot be read (as text) by the computer. *** Documents that were not deemed relevant for the analysis. Documents with the following type: "toespraak", "verantwoording data-analyse", "seminar", "reactie", "protocol", "paper", "panel", "voorstudie", "kwaliteitstoets IOB", "workshop", "lijst met geïnterviewde personen".

## Appendix B

*Appendix B.1. Keywords Used for Search*

**Table A2.** English search terms.

| To combine keywords (Each keyword in the left column is combined with each keyword in the right column, so that the search input is "unintended" AND (effect OR effect of aid OR consequence OR impact OR result OR outcome OR incentive OR benefit OR challenge)) | |
|---|---|
| unintended | effect |
| unintentional | effect of aid |
| unanticipated | consequence |
| unplanned | impact |
| unexpected | result |
| unforeseen | outcome |
| unpredictable | incentive |
| inadvertent | benefit |
| perverse | challenge |
| avoidable | |
| unavoidable | |
| drawback | |
| indirect | |
| undesirable | |
| spillover | |
| disproportionate | |
| unwanted | |
| ripple | |
| boomerang | |
| differential | |
| rebound | |
| recoil | |
| surprise | |
| accidental | |
| **Standalone search terms** | |
| externality | |
| adverse effects | |
| hidden cost | |
| side-effect | |
| malpractice | |
| no harm | |
| synergies | |
| tradeoff/trade-off | |

*Appendix B.2. Dutch Search Terms*

**Table A3.** Dutch search terms.

| To Combine Search Terms | |
| --- | --- |
| onbedoeld | effect |
| onverwacht | gevolg |
| ongepland | impact |
| onvoorzien | uitkomst |
| bijwerking | resultaten |
| ongewild | resultaat |
| onopzettelijk | tekortkoming |
| pervers | consequentie |
| onvermijdelijk | |
| goed bedoeld | |
| indirect | |
| secundair | |
| onwenselijk | |
| ongewenst | |
| negatieve | |
| schadelijk | |
| te vermijden | |
| **Standalone search terms** | |
| keerzijde | |
| spill-over | |
| perverse prikkels | |
| schaduwzijde | |
| tradeoff/trade-off | |
| neveneffecten | |

All keywords and text were stemmed before searching to avoid missing matches due to a different tense or plural. We looked for exact matches in the text using word boundaries. Combined search terms had to be found within one sentence.

## Appendix C. Document Types

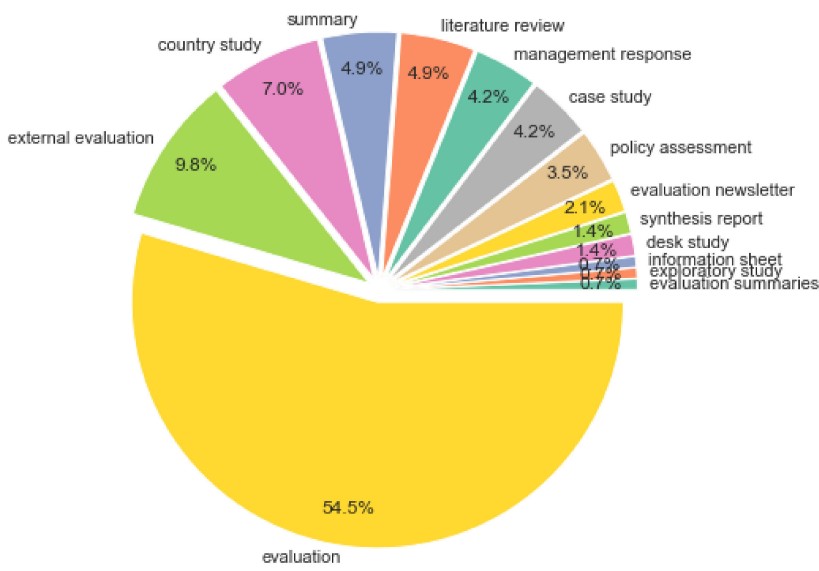

**Figure A1.** Document types of documents reporting upon unintended effects.

## Appendix D. Terms of Reference Documents

The categories of the Terms of Reference documents are as follows:

T1. No search terms.

T2. False positive—no mention of unintended effects in the document.

T3. Unintended effects are only generally mentioned: they mention that in general the evaluation should look into possible unintended or indirect effects.

T4. Specific unintended effects are mentioned: they mention possible, specific unintended effects to which they should pay attention in the evaluation.

**Table A4.** Terms of Reference categories.

| Category | Number of Publications | Share |
|---|---|---|
| T0 | 30 | 33.3% |
| T1 | 29 | 32.2% |
| T2 | 27 | 30.0% |
| T3 | 4 | 4.4% |
| Total | 90 | |

**Table A5.** Examples of the Terms of Reference categories.

| Category | Publication Title | Reason for Category |
|---|---|---|
| T0 | | No search terms found in the document. |
| T1 | Terms of Reference—Evaluation of international trade and investment policy of the Netherlands | The search terms found in this document are about that the potentially indirect impact of international trade policy on economic growth (amongst others) is challenging. There is no instruction that mentions that unintended effects should be taken into account in the evaluation, and there is no evaluation question that covers unintended effects. |
| T2 | Terms of Reference—Evaluation of Dutch Development Cooperation in the Palestinian Territories 2008–2014 | In the ToR the question "What are unanticipated positive or negative consequences of the programme?" is included in the evaluation questions. They do not give concrete examples of possible unintended effects, but they do mention that the evaluation should look into unanticipated consequences. |
| T3 | Terms of Reference—Evaluatie directe financiering van lokale Zuidelijke NGO's door ambassades | The ToR document explicitly notes specific unintended effects that have occurred previously according to analyses (e.g., a shift in attention from target groups to donors, damage of the ownership, etc.). These specific unintended effects are described, and the authors urge evaluators to pay attention to them. |

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
