# Peer review of "Assessing International Development Cooperation: Becoming Intentional about Unintended Effects"

_sustainability, doi:10.3390/su132111571_

Round 1

Reviewer 1 Report

There are a number of things I like about this article. I find it to be highly original, thought provoking, clearly written, and publishable. It addresses a critical but wildly understudied aspect of development cooperation, unintended consequences, and provides a clear typology for grouping those consequences. Furthermore, the authors establish that only a minority of evaluations of development cooperation include assessments of unintended consequences, identifying an oversight on the part of policymakers and practitioners. Nevertheless, the authors compellingly are able to group the identified unintended consequences into a coherent typology. This article should be useful to both policymakers and academics moving forward.

Author Response

Thank you for your encouraging comments. We intend to engage both policy makers and academics, and are pleased to note the considerable interest in initial dissemination efforts.

Reviewer 2 Report

I congratulate the authors for their work and for the exhaustive analysis of the evaluations of their country's international cooperation projects. I also encourage them to continue in the field of research on international development cooperation.

Author Response

Thank you for your encouraging comments. As noted in our response to reviewer 1, we intend engage relevant audiences around a discussion of these findings.

Reviewer 3 Report

The paper entitled” Assessing International Development Cooperation: Becoming Intentional About Unintended Effects” deals with actual and very interesting topic.  I appreciate the aims of this work, the approach followed looks useful and the results are promising.

However, I have the following comments that hopefully help the authors improve their paper:

  • I suggest that the authors add a research method diagram. This will provide a snapshot of the research steps followed and will help the reader in a clearer understanding of the paper.
  • The literature review is insufficient, it must be better contextualized and be more convincing.
  • The authors should convince the readers of this journal, that their contribution is so important. These issues deserve a deeper discussion: What are the managerial implications from this work? How decision or policy makers could benefit from this study.
  • As usual a final thorough proof-reading is recommended.

I encourage the author to think along those questions and to develop this work further along those lines.

Round 2

Reviewer 3 Report

The authors adequately addressed the issues raised in my review